# The Importance of Dose Escalation in the Treatment of Pulmonary Arterial Hypertension with Treprostinil

**DOI:** 10.3390/biomedicines13010172

**Published:** 2025-01-13

**Authors:** Piotr Kędzierski, Marta Banaszkiewicz, Michał Florczyk, Michał Piłka, Rafał Mańczak, Maria Wieteska-Miłek, Piotr Szwed, Krzysztof Kasperowicz, Katarzyna Wrona, Szymon Darocha, Adam Torbicki, Marcin Kurzyna

**Affiliations:** 1Chair and Department of Pulmonary Circulation, Thromboembolic Diseases and Cardiology, Centre of Postgraduate Medical Education, European Health Centre, ERN-LUNG Member, 05-400 Otwock, Poland; piotr.kedzierski@ecz-otwock.pl (P.K.); michal.florczyk@ecz-otwock.pl (M.F.); michal.pilka@ecz-otwock.pl (M.P.); rafal.manczak@ecz-otwock.pl (R.M.); maria.wieteska@ecz-otwock.pl (M.W.-M.); piotr.szwed@ecz-otwock.pl (P.S.); krzysztof.kasperowicz@ecz-otwock.pl (K.K.); katarzyna.wrona@ecz-otwock.pl (K.W.); szymon.darocha@ecz-otwock.pl (S.D.); adam.torbicki@ecz-otwock.pl (A.T.); marcin.kurzyna@ecz-otwock.pl (M.K.); 2Department of Vascular, Endovascular Surgery, Angiology and Phlebology, Poznan University of Medical Science, 61-701 Poznan, Poland

**Keywords:** pulmonary arterial hypertension, prostacyclin, treprostinil, dose escalation, respond to treatment

## Abstract

**Background**: Treprostinil, which is administered via continuous subcutaneous or intravenous infusion, is a medication applied in the treatment of pulmonary arterial hypertension (PAH). The dose of treprostinil is adjusted on an individual basis for each patient. A number of factors determine how well patients respond to treatment. **Objectives**: The aim of this study was to identify factors that may influence the clinical response to the dose of treprostinil at 3 months after the start of therapy. **Methods**: The factors influencing treatment response were analyzed in consecutive PAH patients who started receiving treprostinil treatment. The treatment efficacy was assessed as improvement in 6 min walk distance (6MWD) and WHO functional class (WHO FC), a reduction in N-terminal prohormone of brain natriuretic peptide (NTproBNP), and the percentage of patients achieving low-risk status after 12 months of treatment. **Results**: A total of 83 patients were included in this analysis. Classification of patients according to the tertiles of treprostinil dose achieved at 3 months after drug inclusion shows that after 12 months of follow-up, the median WHO FC in the highest dose group was lower than that in the intermediate dose group (WHO FC II vs. WHO FC III, *p* = 0.005), the median NTproBNP was lower (922 pg/mL, vs. 1686 pg/mL, *p* = 0.036) and 6MWD was longer (300 m vs. 510 m, *p* = 0.015). The French Noninvasive Criteria (NIFC) scale score was higher (2 vs. 0, *p* = 0.008), and the Reveal scale score was lower (5.0 vs. 8.5, *p* = 0.034). In the group of patients who exceeded a dose of 19.8 ng/kg/min within 3 months, an improvement in 6MWD was observed significantly more often after one year of therapy, and they were more likely to show an increase in NIFC scale scores after one year of therapy than the group of patients who received the lower dose (65% vs. 30%, *p* = 0.02). In the group of patients younger than 50 years of age, a statistically significant correlation was observed between the dose of treprostinil achieved after three months of treatment and the parameters assessed after 12 months of treatment, including WHO FC, 6MWD, and NIFC prognostic scale scores (all *p* < 0.05). **Conclusions**: The clinical effect of treatment is critically dependent on the rapid escalation of the treprostinil dose during the first three months of treatment.

## 1. Introduction

Pulmonary arterial hypertension (PAH) is a rare and chronically progressive disease. If left untreated, it can lead to right ventricular (RV) failure and death [1]. PAH mortality in Poland in 2021 was 0.1–0.2 deaths per 100,000 people [2]. The pathophysiology of the disease involves progressive narrowing of the lumen of the pulmonary arteries, leading to increased pulmonary vascular resistance and increased pressure in the pulmonary circulation. In recent years, several medications have been developed that act through a variety of mechanisms to inhibit pulmonary vascular remodeling and increase the diameter of pulmonary vessels. There are four main groups of medications used in clinical practice: orally administered phosphodiesterase 5 inhibitors (PDE5i: sildenafil daily dose 60 mg [3], tadalafil daily dose 40 mg [4]), endothelin receptor antagonists (ERA: bosentan daily dose 250 mg [5], macitentan daily dose 10 mg [6], ambrisentan daily dose 10 mg [7]), and parenterally or orally administered prostacyclins, their analogs or receptor agonists (treprostinil administered as a continuous subcutaneous or intravenous infusion, through inhalation or as an oral form [8] in doses depending on the clinical effect of treatment and medication tolerability, epoprosenol administered as an intravenous continous infusion in doses depending on the effect and tolerability [9], iloprost administrated as an inhalation in a maximum daily dose of 45 μg [10], selexipag by oral administration, maximum daily dose of 2 × 1600 μg [11]), and activin signaling inhibitors (sotatercept administrated as subcutaneous injection, target dose 0.7 mg/kg every 3 weeks [12]). Their action in the treatment of PAH is through vasodilatation—PDE5i, ERA, prostacyclins, and the inhibition of smooth muscle cell proliferation—sotatercept [8,13]. Treprostinil is a synthetic derivative of prostacyclin administered through continuous subcutaneous infusion, intravenously, or via inhalation [14]. The target medication dose has been determined for the inhaled route of administration (nine breaths—54 mcg—per treatment session four times a day), which is used primarily in patients with pulmonary hypertension associated with interstitial lung disease [15]. For subcutaneous, intravenous, and oral administration (dosing three times a day), target doses of the medication have not been determined and depend on the clinical effect and treatment tolerance [16,17]. An important clinical issue is the dose escalation of treprostinil, which is limited by the side effects of the medication related to its activity or the subcutaneous mode of administration. Clinical practice shows that some patients achieve significant clinical improvement in response to treatment, while in other patients, the treatment effect is not directly related to the dose. The aim of this study was to analyze the factors that impact the clinical effectiveness of treprostinil in relation to the dose achieved at 3 months.

## 2. Materials and Methods

### 2.1. Selection of Patients and Data Recording

This is a retrospective observational study that aims to assess a group of patients treated with treprostinil (Tresuvi^®^ (Amomed, Vienna, Austria), Remodulin^®^ (Ferrer International, Barcelona, Spain)) for PAH as an addition to existing PAH-targeted therapy or as up-front therapy between 2012 and 2022. The diagnosis of PAH was made in accordance with current ESC guidelines [14]. According to the principles of reimbursement policy for treprostinil in Poland during this study, the drug was administered to patients who failed to achieve therapeutic goals despite oral targeted therapy (PDE5i and ERA) and as up-front combination therapy in patients in WHO functional class (WHO FC) IV at the time of diagnosis. In the study group of patients, the mean time from the diagnosis of PAH to the start of treprostinil treatment was 2400 days (±6832), with a median of 625 days (IQR 234-2255).

Prior to the start of treprostinil treatment, all patients underwent a noninvasive evaluation with the following risk assessment systems: Compera 2 [18], French Noninvasive Criteria [19], and Reveal lite 2 [20]. Right heart catheterization (RHC) was also performed according to current standards [21]. Patients were assessed at the 12-month mark following the start of therapy with treprostinil.

The response to treatment was defined in line with the following criteria: a decrease in the WHO functional class, an extension of the 6 min walk distance (6MWD), a decrease in the serum concentration of the N-terminal prohormone of brain natriuretic peptide (NTproBNP), a decrease in the score on the Compera 2 and Reveal lite 2 prognostic scales, and an increase in the French Noninvasive Criteria (NIFC) score. The response to treatment was analyzed based on the patient’s age at the time of treprostinil administration and the dosage achieved. The initial analysis involved the classification of patients into tertiles based on the dose of the medication they had received after 3 months of treatment. The second analysis included patients who reached a treprostinil dose of 19.8 ng/kg/min or higher (the dose corresponding to the third tertile when the patient group was divided based on the drug dose achieved after 3 months of treatment) after 3 months of treatment and a subgroup in which the drug dose was lower. The third analysis investigated the age of the patients at the time of their inclusion in the study group, classified as below or above 50 years of age (the median age in the study group was 49 years).

### 2.2. Statistical Analysis

A statistical analysis was performed using the STATISTICA software (v. 13, Oklahoma City, OK, USA). The distribution of data was verified using the Shapiro–Wilk test. Continuous variables are presented as median and interquartile range (IQR), and nominal variables are presented as numbers and percentages. The Mann–Whitney U-test was used to compare continuous variables between two groups, while the Kruskal–Wallis test was used to make comparisons between three groups. The chi-squared test was used for nominal variable comparisons. Correlations between variables were tested by determining Spearman’s correlation coefficient. The statistical significance level was *p* < 0.05.

## 3. Results

### 3.1. Population Characteristics

A total of 83 consecutive patients with PAH who received treprostinil through continuous subcutaneous or intravenous infusion at the referral center between 2012 and 2022 were included in the analysis (Table 1). The study population included 51 women (61%) and 32 men (39%). A total of 58 patients (70%) were diagnosed with idiopathic PAH, while 13 patients (16%) had PAH associated with connective tissue disease (CTD). The remaining 10 patients (12%) had PAH associated with congenital heart defects, and 2 patients (2%) had portopulmonary hypertension.

Before starting treatment with treprostinil, 30 patients (36%) were in WHO functional class IV, 45 patients (54%) were in functional class III, and 8 patients (10%) were in functional class II. Prior to the inclusion of treprostinil treatment, the mean arterial pressure (mPAP) was 58 ± 16 mmHg, mean right atrial pressure (RAP) was 9 ± 4 mmHg, cardiac index (CI) was 2.2 ± 0.5 L/min/m^2^, pulmonary vascular resistance (PVR) was 12.9 ± 4.6 Wood units, and mixed venous blood saturation (mvsat) was 67.3 ± 6.8%. Prior to the start of treprostinil treatment, five patients were treatment-naïve and received up-front triple combination therapy (treprostinil + PDE5i +ERA). Of the remaining patients, 26 were taking one medication (PDE5i), 50 were taking two medications (PDE5i + ERA), and 2 were receiving three medications (PDE5i + ERA + inhaled iloprost). The median age of the patients included in the study group was 49 years.

The dose escalation regimen we adopted involved the start of treatment with a dose of 1.25 ng/kg/min, followed by an escalation by 1.25 ng/kg/min twice daily for 4 consecutive days, resulting in a dose of 10 ng/kg/min after 4 days, at which point the patient was discharged home. Following a 7-day interval, the dosage was increased by 2 ng/kg/min every 5 days until a dose of 20 ng/kg/min was reached, which was achieved within 28 days of the start of treatment with treprostinil. Subsequent escalations by 1.25 ng/kg/min every 7 days allowed the intended dose of 30 ng/kg/min to be achieved within 3 months of the start of the treatment. The dosage regimen outlined above was individually adjusted for each patient, depending on treatment tolerance.

After 12 months, 58 patients (70%) continued treatment, while 25 patients (30%) discontinued treatment due to death (16 patients—19%). Intolerance to the treatment manifested as headache, jaw pain, or infusion site pain (five patients—6%); the deterioration of PAH and a subsequent transition to epoprostenol (three patients—3.6%); and lung transplantation (one patient—1%). These results are presented in Figure 1. The average drug doses achieved were as follows: after 6 months, 27.3 ng/kg/min (median 23.5 ng/kg/min); after 9 months, 34.2 ng/kg/min (median 34.9 ng/kg/min); and after 12 months, 40.9 ng/kg/min (median 40.4 ng/kg/min).

Before receiving treatment with treprostinil, 1 patient (1%) was in the low-risk group, 18 patients (22%) were in the intermediate- to low-risk group, 33 patients (40%) were in the intermediate- to high-risk group, and 31 patients (37%) were in the high-risk group, according to the 2022 ESC/ERS guidelines. After 12 months of treatment, 12 patients (20%) were in the low-risk group, 15 patients (26%) were in the intermediate- to low-risk group, 22 (37%) patients were in the intermediate- to high-risk group, and 10 patients (17%) were in the high-risk group (Figure 2).

The achievement of the low-risk profile after 12 months in the analyzed subgroups is shown in the diagram below (Figure 3).

### 3.2. Dose After 3 Months—Split into Terciles

After 12 months of treatment, a significant improvement (*p* < 0.05) in the WHO functional class, a decrease in the NTproBNP levels, an increase in 6MWT distance, an increase in the NIFC prognostic score, and a decrease in the Reveal lite 2 prognostic score were observed only in the group of patients who reached the highest treprostinil dose (>19.8 ng/kg/min) when the population was divided into tertiles (Table 2).

### 3.3. “Rapid” Dose Escalation Scheme Versus “Slow” Dose Escalation

An increase in 6MWT after 12 months of treatment was observed in 87% of patients in the subgroup with the higher dose (>19.8 ng/kg/min—“rapid”) after 3 months versus an increase in 50% of patients in the subgroup with the lower treprostinil dose after 3 months (≤19.8 ng/kg/min—“slow”; *p* = 0.018).

Moreover, an absolute value of 6MWT above 440 m after 12 months of treatment was observed in 67% of patients in the subgroup with the higher dose, while only 31% of patients in the lower-dose subgroup achieved an absolute value of 6MWT above 440 m (*p* = 0.024). Improvements were also observed in terms of a reduction in the WHO functional class, a decrease in NTproBNP concentration (Figure 4), and the achievement of an absolute value of 320 m in 6MWT. However, these parameters did not meet the statistical significance criterion. In addition, the group of patients who exceeded the 19.8 ng/kg/min dose within 3 months was significantly more likely to see an increase in NIFC scale scores after one year of therapy than patients in the lower-dose group (65% vs. 30%; *p* = 0.02) (Figure 5).

### 3.4. Age-Dependent Response to Treatment

In the group of patients younger under 50 years of age, a statistically significant correlation was observed between the treprostinil dose achieved after 3 months of treatment and the parameters assessed after 12 months of treatment; the higher the treprostinil dose was after 3 months of treatment, the lower the WHO grade assessed at 12 months, the longer the 6MWT distance, and the higher the NIFC prognostic score (Table 3). This correlation was not present in the subpopulation of patients over the age of 50.

## 4. Discussion

The results of our study indicate that treprostinil dose escalation during the first 3 months of treatment and the dose achieved during this period are crucial. Reaching a dose of about 20 ng/kg/min within 12 weeks of initiating treprostinil treatment is more effective in improving exercise capacity and right ventricular function after 12 months of treprostinil therapy.

Achieving a dose of 20 ng/kg/min after 3 months may prove challenging due to typical adverse effects of the medication, mainly site pain. This dose significantly exceeds the dose recommended in a pioneering study [22], which recommended an initial dose of 1.25 ng/kg/min, with dose increases of no more than 1.25 ng/kg/min for the first 4 weeks and then no more than 2.5 ng/kg/min per week. This resulted in an average dose of 9.6 ng/kg/min, which is below the optimal level. This method of dose escalation should be considered inadequate today. Initially, it appeared that the incidence of pain at the site of treprostinil administration was dose- and rate-dependent, but subsequent studies showed no such correlation [22,23]. Pain at the site of subcutaneous drug administration and accompanying swelling can be well controlled with cold compresses, nonsteroidal anti-inflammatory drugs, or short courses of high-dose oral prednisolone (2 mg/kg/d) [24]. In a prospective study showing the efficacy and safety of rapid treprostinil dose escalation [25], which was defined as reaching a dose of 30 ng/kg/min 12 weeks after the drug was introduced, the authors also applied opioid drugs to negate local pain. In the study, 3 out of 40 patients (7.5%) discontinued treatment at the 16-week follow-up due to local inflammatory reactions. In our analysis, 5 out of 83 patients (6%) discontinued treatment due to local skin reactions during the 12-month follow-up period. The analgesic treatments we used were nonsteroidal anti-inflammatory drugs, local anesthetics, and cooling compresses. In cases of particularly severe inflammatory reactions at the site of subcutaneous administration of the drug that were causing severe pain and local complications in the form of infection of the subcutaneous tissue, the problem was solved by implanting a subcutaneous pump to deliver treprostinil intravenously. Although the administration of the drug through the implantable pump is not free of complications—primarily local complications in the form of hematoma and wound infection, as well as the dislocation of the catheter that delivers the drug to the bloodstream and the acceleration of drug delivery observed over time [26]—it significantly improves the quality of life of patients with local skin reactions [27].

The beneficial effect of rapid dose escalation was confirmed in a study of another prostacyclin, epoprostenol, by Tokunaga et al. [28]. During a two-year follow-up of 46 patients, significantly higher doses of epoprostenol were observed in the first months of escalation in the survivor group than in the non-survivor group. In the rapid dose escalation subpopulation (>20 ng/kg/min in 3 months and 45 ng/kg/min in 12 months), there was a greater decrease in mPAP and a noticeably better 9.5-year survival rate (100% vs. 64%, *p* = 0.022).

The effectiveness of rapid escalation may be due to the progressive nature of pulmonary arterial hypertension. The earlier inhibition of changes that result in pulmonary vascular remodeling allows for slowing disease progression at an earlier stage. Similar results were obtained by Skoro-Sajer N. et al. [23] in a prospective comparison of two groups of patients. In the slow escalation group, the average dose of treprostinil after 3 months of treatment was 12.9 ng/kg/min, while in the rapid escalation group, it was 20.3 ng/kg/min. In the rapid escalation group, a statistically significantly higher increase in the 6MWT distance was observed compared to the slow escalation group. The results of our study also indicate a beneficial effect of rapid dose escalation of treprostinil on 6MWT improvement. In the rapid dose escalation group (>20 ng/kg/min at 3 months), improvement in 6MWT was observed in 87% of patients after 12 months of treatment, while in the slow escalation group, such a benefit was observed in 50% of patients (*p* = 0.018). In the rapid treprostinil dose escalation group, after 12 months of treatment, an absolute 6MWT distance > 440 m was achieved by 67% of patients, while in the slow escalation group, it was achieved by only 31% of patients (*p* = 0.024).

An important question is how high the treprostinil dose should be escalated to achieve the optimal clinical effect. The maximum dose of the drug has been specified by the manufacturer for inhaled administration only [29]. When administered subcutaneously and intravenously, the dose should be escalated until symptoms are reduced and according to patient tolerance. There are no data in the literature that specify what dose of the drug offers an optimal compromise. In our study, we were also unable to conclusively demonstrate that a specific dose level was associated with improvements in the prognostic parameters of PAH treatment after 12 months of therapy. The minimum dose after 12 months of treatment in the patient group in our study was 13.2 ng/kg/min, while the maximum dose was 86 ng/kg/min.

A different perspective on the impact of the time from the diagnosis of PAH to the initiation of optimal treatment is provided by the work of Papa S. et al. [30]. They found that the inclusion of up-front triple-targeted therapy (PDE5i, ERA, and treprostinil) provides a greater reduction in PVR and allows for achieving a low-risk profile among a larger number of PAH patients than the addition of treprostinil to double oral therapy (add-on strategy: PDE5i and ERA, then the addition of prostanoid). The results of our study indicate an important role in the rate of treprostinil dose escalation during the first 3 months after treatment is started. The analysis of the results of both studies indicates that in the treatment of PAH, it is crucial to implement combined therapy as soon as possible and to escalate the doses of targeted drugs as quickly as possible. Moreover, D’Alto et al. [31] showed that starting PAH treatment with triple therapy (PDE5i, ERA, treprostinil) in newly diagnosed patients results in clinical improvement (WHO functional class decreased from 3.4 ± 0.5 to 2.0 ± 0.4, and 6MWT increased from 158 ± 130 to 431 ± 66 m (both *p* < 0.001)) and hemodynamic improvement (CI increased from 1.8 ± 0.3 to 3.5 ± 0.8 L/min/m^2^, *p* < 0.001) and allows for right-sided cardiac remodeling (decreased right atrial and RV areas, improved left ventricular eccentricity index and increased fractional area change (all *p* < 0.001)).

The demonstration of a correlation between the dose of treprostinil achieved after 3 months of treatment and the improvement in the 6MWT, WHO class, and NIFC prognostic score after 12 months of treatment in patients less than 50 years of age indicates a better clinical response in younger patients. This is most likely due to the absence of additional chronic diseases that affect the general condition of patients and lead to a worse response to treatment. Jonathan A Rose et al. [32] described differences among the PAH patient population according to age. The group of 2627 patients was divided into subgroups according to age: <50 years of age, 51–64 years of age, and ≥65 years of age. The oldest group of patients (compared to the youngest group) had the highest proportion of patients with concomitant connective tissue disease, patients in WHO functional classes III and IV, and patients who exhibited the shortest distance on the walk test and lower mean pulmonary pressure during RHC upon PAH diagnosis. These findings indicate that in older patients, PAH is often an additional disease that is one of the factors limiting exercise tolerance. In this case, including a targeted treatment for PAH may yield relatively worse results because we are only reducing one of the disease factors. Similar results to our study were shown by Hoeper et al. [33], who analyzed the COMPERA registry. In that study, the group of patients over 65 years of age achieved less improvement in the 6MWT after 3 and 12 months of targeted PAH treatment, and the proportion of patients with 6MWT > 400 m was lower in elderly patients than in younger patients.

## 5. Limitations

Our study has several limitations, the most important being its retrospective, noncontrolled, single-center nature. In addition, for reimbursement reasons and payer requirements, a very small number of patients have been able to receive triple up-front therapy, which currently appears to have the best clinical results.

## 6. Conclusions

In conclusion, it should be emphasized that both our results and those of other studies indicate that the inclusion of treprostinil in treatment should not be delayed. Early treatment initiation and rapid dose escalation, together with control of the drug side effects and in combination with the other groups of targeted therapies (PDE5i and ERA), offer a chance to slow down disease progression and improve the clinical status of patients.

## Figures and Tables

**Figure 1 biomedicines-13-00172-f001:**
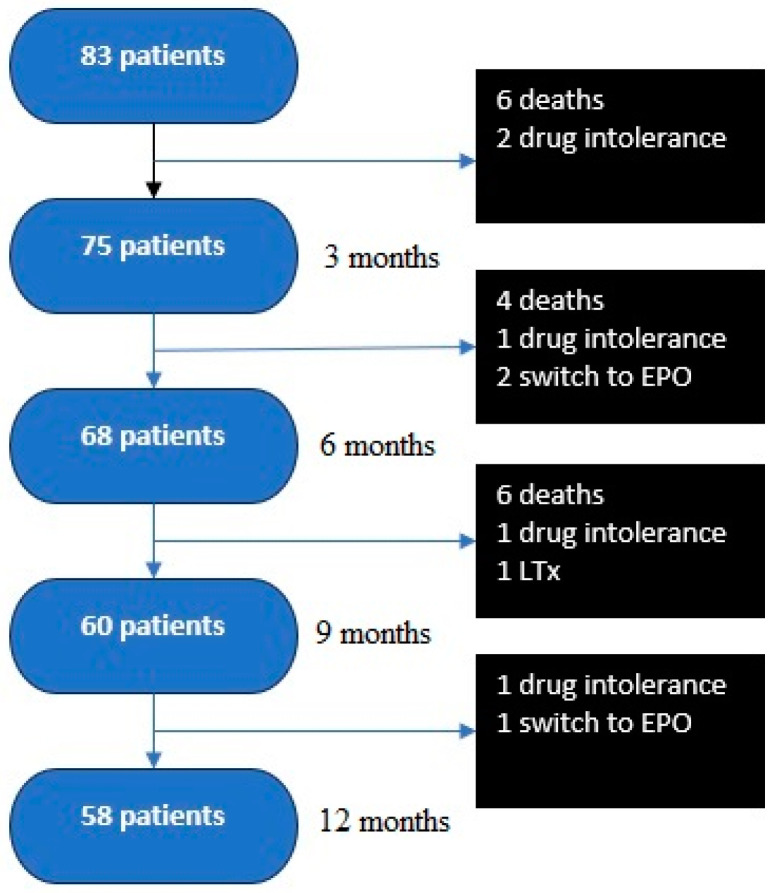
Reasons for the discontinuation of treprostinil treatment within the first 12 months of treatment. EPO—epoprostenol; LTx—lung transplantation.

**Figure 2 biomedicines-13-00172-f002:**
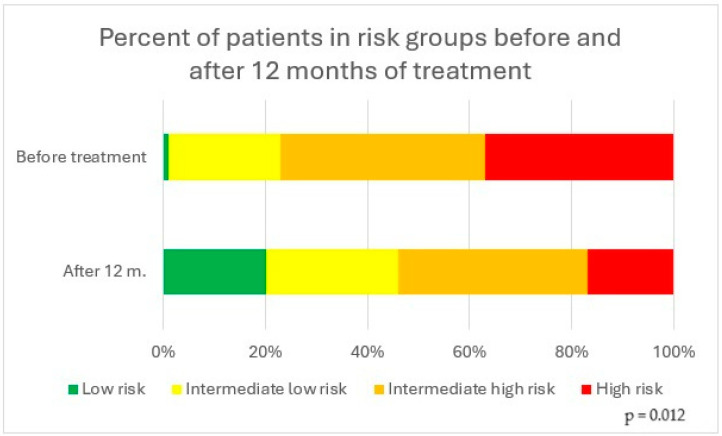
Percentage of patients in risk groups before and after 12 months of treatment.

**Figure 3 biomedicines-13-00172-f003:**
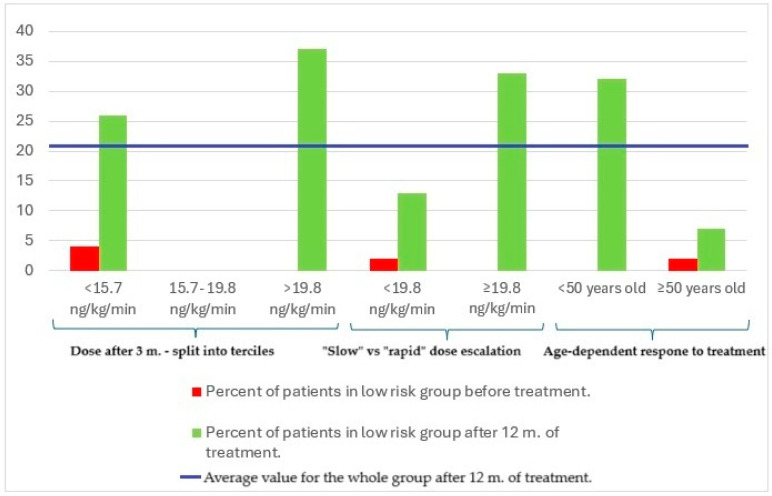
Percentage of patients in the low-risk group before treatment and after 12 months of treatment.

**Figure 4 biomedicines-13-00172-f004:**
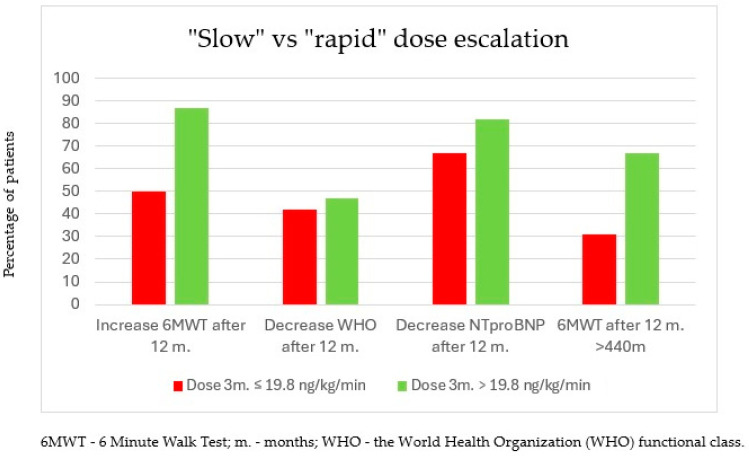
Improvement in functional and biochemical parameters after 12 months of treatment in subgroups of patients who achieved a dose of treprostinil >19.8 ng/kg/min vs. those who received a dose ≤19.8 ng/kg/min after 3 months of treatment.

**Figure 5 biomedicines-13-00172-f005:**
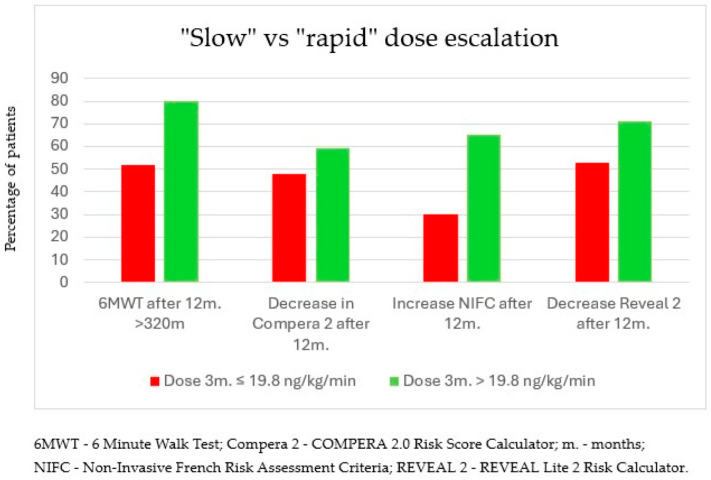
Improvement in functional and biochemical parameters after 12 months of treatment in subgroups of patients who achieved a dose of treprostinil >19.8 ng/kg/min vs. those who received a dose ≤19.8 ng/kg/min after 3 months of treatment.

**Table 1 biomedicines-13-00172-t001:** Baseline characteristics.

Variables	All Patients	<19.8 ng/kg/min in 3 m	≥19.8 ng/kg/min in 3 m	<50 Years Old	≥50 Years Old
Clinical data	n = 83	n = 50	n = 25	n = 35	n = 48
Sex					
Male	32	20	9	14	18
Female	51	30	16	21	30
WHO FC					
I	0	0	0	0	0
II	8 (10%)	3 (6%)	5 (20%)	6 (17%)	2 (4%)
III	45 (54%)	30 (60%)	11 (44%)	24 (69%)	21 (44%)
IV	30 (36%)	17 (34%)	9 (36%)	5 (14%)	25 (52%
Median (IQR) 6MWD (m)	237 (0–450)	302 (0–442)	397 (31–484)	417 (299–469.5)	141 (0–299.5)
Median (IQR) NTproBNP (pg/mL)	2006 (946–3958.5)	2232 (797–3908)	1923 (1139–3239)	1239 (546.5–2576.5)	2826 (1358.5–6615.5)
Median (IQR) Prognostic scale					
Compera 2	3 (3–4)	3 (3–4)	3 (2–4)	3 (2–3)	4 (3–4)
NIFC	0 (0–1)	0 (0–1)	0 (0–1)	1 (0–1)	0 (0–0)
REVEAL 2	9 (6–11)	9 (6–11)	9 (6–11)	7 (5.5–9)	11 (8–12)

6MWD—6 min walk distance; Compera 2—COMPERA 2.0 Risk Score; m—months; NIFC—Noninvasive French Risk Assessment Criteria; REVEAL 2—REVEAL Lite 2 Risk Score; WHO FC—World Health Organization (WHO) functional class.

**Table 2 biomedicines-13-00172-t002:** Division of patients according to dose level after 3 months into terciles—parameteres after 12 months of treatment.

	Dose < 15.7 ng/kg/min (n = 24) Median (IQR)	Dose 15.7–19.8 ng/kg/min (n = 26) Median (IQR)	Dose > 19.8 ng/kg/min (n = 26) Median (IQR)	*p*
WHO FC after 12 m.	3 (2–3)	3 (3–3)	2 (2–3)	0.005
NTproBNP after 12 m.	535 (280–1247)	1686 (1167–2294)	922 (221–1435)	0.036
6 MWD after 12 m.	392 (282–453)	300 (217–450)	510 (408–531)	0.015
Compera 2 after 12 m.	2.5 (1.5–3.0)	3 (2–3)	2 (1.0–3.0)	0.069
NIFC after 12 m.	1 (0–2)	0 (0–1)	2 (1–3)	0.008
REVEAL 2 after 12 m.	6.5 (4.0–10.0)	8.5 (6.0–11.0)	5.0 (3.0–8.0)	0.034

6MWD—6 min walk distance; Compera 2—COMPERA 2.0 Risk Score; m.—months; NIFC—Non-invasive French Risk Assessment Criteria; NTproBNP—N-terminal prohormone of brain natriuretic peptide; REVEAL 2—REVEAL Lite 2 Risk; WHO FC—The World Health Organization (WHO) functional class.

**Table 3 biomedicines-13-00172-t003:** Spearman’s correlation coefficients for treprostinil dose achieved in subsequent escalation stages and clinical parameters.

	WHO FC After 12 m	NTproBNP After 12 m	6MWD After 12 m	Compera After 12 m	NIFC After 12 m	Reveal After 12 m
Dose 3 m	−0.525	−0.052	0.501	−0.241	0.393	−0.352
Dose 6 m	−0.316	−0.065	0.315	−0.043	0.201	−0.168
Dose 9 m	−0.049	0.040	0.208	0.102	0.037	−0.060
Dose 12 m	−0.270	−0.016	0.070	−0.348	0.084	−0.094

6MWD—6 min walk distance; Compera 2—COMPERA 2.0 Risk Score; m.—months; NIFC—Non-invasive French Risk Assessment Criteria; NTproBNP—N-terminal prohormone of brain natriuretic peptide; REVEAL 2—REVEAL Lite 2 Risk; WHO FC—The World Health Organization (WHO) functional class.

## Data Availability

Data are unavailable due to privacy restrictions.

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
