# Peer review of "The Importance of Dose Escalation in the Treatment of Pulmonary Arterial Hypertension with Treprostinil"

_biomedicines, 2025, doi:10.3390/biomedicines13010172_

Round 1
Reviewer 1 Report
Comments and Suggestions for Authors
Authors assessed various parameters such as 6-minute walk distance (6MWD), WHO functional class (WHO FC), a reduction in N-terminal prohormone of brain natriuretic peptide (NTproBNP) and percentage of patients scoring low-risk status involving 83 patients. Different criteria set for the study and in patients younger than 50 years of age, authors obtained a statistically significant correlation (WHO FC, 6MWD and NIFC prognostic scale scores). Authors finally concluded that the clinical effect of treatment is critically dependent on the rapid escalation of the dose of Treprostinil during the first 3 months. This is one of the interesting studies reported by the authors to study the time frame of the therapy of PAH. The obtained results were discussed and the conclusion is aligned to the objective of the study.
Following minor comments need to be addressed to improve the contents of the manuscript:-
1. Increase the number of keywords as per the journal guidelines.
2. Introduction section:-
(a) Include a recent statistical data related to the mortality rate of patients suffered with PAH/RV failure deaths in the specific country as per WHO reports.
(b) Authors need to give an account on the doses and route of administration available four main groups of medications with citations, respectively.
(c) Reference number 2 is published 4 years ago, any recent literature pertaining to the Treprostinil to be cited and include more details about this drug, the prescribed/recommended dose level in three types of route of administration with recent literature.
(d) After citing reference number 2, there is no citation of the relevant literature, hence, authors need to describe the last paragraph and include more literature references, recently published ones.
3. Materials and Methods:-
(a). Authors need to mention the brand name of the medication used to the patients.
(b) The key reference (ref-4) has published in the year 2015/2016 (ESC guidelines). Authors mentioned the data collected between 2012-2022 (a period of decade). It seems in European Union, the drug was approved in the year 2020. Authors need to justify this point.
Author Response
- Increase the number of keywords as per the journal guidelines.
Authors: Thank You for this comment. The number of keywords has been modified in accordance with your suggestion.
Keywords: pulmonary arterial hypertension; prostacyclin; treprostinil; dose escalation; respond to treatment.
- Introduction section:
(a) Include a recent statistical data related to the mortality rate of patients suffered with PAH/RV failure deaths in the specific country as per WHO reports.
Authors: Thank You for this comment. The information has been added according to your suggestion.
PAH mortality in Poland in 2021 was 0.1-0.2 deaths per 100,000 population [2].
- GBD 2021 Pulmonary Arterial Hypertension Collaborators. Global, regional, and national burden of pulmonary arterial hypertension, 1990-2021: a systematic analysis for the Global Burden of Disease Study 2021. Lancet Respir Med. 2024 Oct 18:S2213-2600(24)00295-9. doi: 10.1016/S2213-2600(24)00295-9. Epub ahead of print. PMID: 39433052.
(b) Authors need to give an account on the doses and route of administration available four main groups of medications with citations, respectively.
Authors: Thank You for this comment. The information has been added according to your suggestion.
There are four main groups of medications used in clinical practice: orally administered phosphodiesterase 5 inhibitors (PDE5i: sildenafil daily dose 60mg [1], tadalafil daily dose 40mg [2]), endothelin receptor antagonists (ERA: bosentan daily dose 250mg [3], macitentan daily dose 10mg [4], ambrisentan daily dose 10mg [5]), and parenterally or orally administered prostacyclins, their analogs or receptor agonists (treprostinil administered as a continuous subcutaneous or intravenous infusion, by inhalation or as an oral form [6] in doses depending on the clinical effect of treatment and medication tolerability, epoprosenol administered as an intravenous continous infusion in doses depending on the effect and tolerability [7], iloprost administrated as an inhalation in a maximum daily dose of 45ug [8], selexipag by oral administration, maximum daily dose of 2 x 1600ug [9]) and activin signaling inhibitor (sotatercept administrated as subcutaneous injection, target dose 0. 7 mg/kg every 3 weeks [10]).
- Rubin LJ, Badesch DB, Fleming TR, Galiè N, Simonneau G, Ghofrani HA, Oakes M, Layton G, Serdarevic-Pehar M, McLaughlin VV, Barst RJ; SUPER-2 Study Group. Long-term treatment with sildenafil citrate in pulmonary arterial hypertension: the SUPER-2 study. Chest. 2011 Nov;140(5):1274-1283. doi: 10.1378/chest.10-0969. Epub 2011 May 5. PMID: 21546436.
- Galiè N, Brundage BH, Ghofrani HA, Oudiz RJ, Simonneau G, Safdar Z, Shapiro S, White RJ, Chan M, Beardsworth A, Frumkin L, Barst RJ; Pulmonary Arterial Hypertension and Response to Tadalafil (PHIRST) Study Group. Tadalafil therapy for pulmonary arterial hypertension. Circulation. 2009 Jun 9;119(22):2894-903. doi: 10.1161/CIRCULATIONAHA.108.839274. Epub 2009 May 26. Erratum in: Circulation. 2011 Sep 6;124(10):e279. Dosage error in article text. PMID: 19470885.
- Channick RN, Simonneau G, Sitbon O, Robbins IM, Frost A, Tapson VF, Badesch DB, Roux S, Rainisio M, Bodin F, Rubin LJ. Effects of the dual endothelin-receptor antagonist bosentan in patients with pulmonary hypertension: a randomised placebo-controlled study. Lancet. 2001 Oct 6;358(9288):1119-23. doi: 10.1016/S0140-6736(01)06250-X. PMID: 11597664.
- Pulido T, Adzerikho I, Channick RN, Delcroix M, Galiè N, Ghofrani HA, Jansa P, Jing ZC, Le Brun FO, Mehta S, Mittelholzer CM, Perchenet L, Sastry BK, Sitbon O, Souza R, Torbicki A, Zeng X, Rubin LJ, Simonneau G; SERAPHIN Investigators. Macitentan and morbidity and mortality in pulmonary arterial hypertension. N Engl J Med. 2013 Aug 29;369(9):809-18. doi: 10.1056/NEJMoa1213917. PMID: 23984728.
- Galiè N, Olschewski H, Oudiz RJ, Torres F, Frost A, Ghofrani HA, Badesch DB, McGoon MD, McLaughlin VV, Roecker EB, Gerber MJ, Dufton C, Wiens BL, Rubin LJ; Ambrisentan in Pulmonary Arterial Hypertension, Randomized, Double-Blind, Placebo-Controlled, Multicenter, Efficacy Studies (ARIES) Group. Ambrisentan for the treatment of pulmonary arterial hypertension: results of the ambrisentan in pulmonary arterial hypertension, randomized, double-blind, placebo-controlled, multicenter, efficacy (ARIES) study 1 and 2. Circulation. 2008 Jun 10;117(23):3010-9. doi: 10.1161/CIRCULATIONAHA.107.742510. Epub 2008 May 27. PMID: 18506008.
- Feldman J, Habib N, Fann J, Radosevich JJ. Treprostinil in the treatment of pulmonary arterial hypertension. Future Cardiol. 2020 Nov;16(6):547-558. doi: 10.2217/fca-2020-0021. Epub 2020 May 11. PMID: 32391733.
- Barst RJ, Rubin LJ, Long WA, McGoon MD, Rich S, Badesch DB, Groves BM, Tapson VF, Bourge RC, Brundage BH, Koerner SK, Langleben D, Keller CA, Murali S, Uretsky BF, Clayton LM, Jöbsis MM, Blackburn SD, Shortino D, Crow JW; Primary Pulmonary Hypertension Study Group. A comparison of continuous intravenous epoprostenol (prostacyclin) with conventional therapy for primary pulmonary hypertension. N Engl J Med. 1996 Feb 1;334(5):296-301. doi: 10.1056/NEJM199602013340504. PMID: 8532025.
- Ewert R, Gläser S, Bollmann T, Schäper C. Inhaled iloprost for therapy in pulmonary arterial hypertension. Expert Rev Respir Med. 2011 Apr;5(2):145-52. doi: 10.1586/ers.11.14. PMID: 21510725.
- Sitbon O, Channick R, Chin KM, Frey A, Gaine S, Galiè N, Ghofrani HA, Hoeper MM, Lang IM, Preiss R, Rubin LJ, Di Scala L, Tapson V, Adzerikho I, Liu J, Moiseeva O, Zeng X, Simonneau G, McLaughlin VV; GRIPHON Investigators. Selexipag for the Treatment of Pulmonary Arterial Hypertension. N Engl J Med. 2015 Dec 24;373(26):2522-33. doi: 10.1056/NEJMoa1503184. PMID: 26699168.
- Hoeper MM, Badesch DB, Ghofrani HA, Gibbs JSR, Gomberg-Maitland M, McLaughlin VV, Preston IR, Souza R, Waxman AB, Grünig E, Kopeć G, Meyer G, Olsson KM, Rosenkranz S, Xu Y, Miller B, Fowler M, Butler J, Koglin J, de Oliveira Pena J, Humbert M; STELLAR Trial Investigators. Phase 3 Trial of Sotatercept for Treatment of Pulmonary Arterial Hypertension. N Engl J Med. 2023 Apr 20;388(16):1478-1490. doi: 10.1056/NEJMoa2213558. Epub 2023 Mar 6. PMID: 36877098.
(c) Reference number 2 is published 4 years ago, any recent literature pertaining to the Treprostinil to be cited and include more details about this drug, the prescribed/recommended dose level in three types of route of administration with recent literature.
Authors: Thank You for this comment. The information has been completed according to your suggestion.
The target medication dose has been determined for the inhaled route of administration (9 breaths - 54 mcg - per treatment session 4 times a day), which is used especially in patients with pulmonary hypertension associated with interstitial lung disease [16]. For subcutaneous, intravenous and oral administration (dosing 3 times a day), target doses of the medication have not been determined, and depend on the clinical effect and tolerance of treatment [17,18].
- Nathan SD, Deng C, King CS, DuBrock HM, Elwing J, Rajagopal S, Rischard F, Sahay S, Broderick M, Shen E, Smith P, Tapson VF, Waxman AB. Inhaled Treprostinil Dosage in Pulmonary Hypertension Associated With Interstitial Lung Disease and Its Effects on Clinical Outcomes. Chest. 2023 Feb;163(2):398-406. doi: 10.1016/j.chest.2022.09.007. Epub 2022 Sep 15. PMID: 36115497; PMCID: PMC10083130.
- Balasubramanian VP, Safdar Z, Sketch MR, Broderick M, Nelsen AC, Lee D, treprostinil in the outpatient setting. Pulm Circ. 2022 Jan 12;12(1):e12016. doi: 10.1002/pul2.12016. PMID: 35506102; PMCID: PMC9052964.
- Grünig E, Rahaghi F, Elwing J, Vizza CD, Pepke-Zaba J, Shen J, Yao H, Hage A, Rosenkranz S, Vonk M, Balasubramanian V, Yuanhua Y, Yu Z, Lordan J, Cadaret L, Grover R, Ousmanou A, Seaman S, Deng C, Broderick M, White RJ; FREEDOM-EV Investigators. Oral Treprostinil is Associated with Improved Survival in FREEDOM-EV and its Open-Label Extension. Adv Ther. 2024 Feb;41(2):618-637. doi: 10.1007/s12325-023-02711-x. Epub 2023 Dec 6. PMID: 38055186; PMCID: PMC10838815.
(d) After citing reference number 2, there is no citation of the relevant literature, hence, authors need to describe the last paragraph and include more literature references, recently published ones.
Authors: Thank You for this comment. The information has been completed according to your suggestion together with the previous subsection (c).
- Materials and Methods:
- Authors need to mention the brand name of the medication used to the patients.
Authors: Thank You for this comment. The information has been completed according to your suggestion.
This is a retrospective observational study that aims to assess a group of patients treated with treprostinil (Tresuvi®, Remodulin®) for PAH as an addition to existing PAH-targeted therapy or as up-front therapy between 2012 and 2022.
- The key reference (ref-4) has published in the year 2015/2016 (ESC guidelines). Authors mentioned the data collected between 2012-2022 (a period of decade). It seems in European Union, the drug was approved in the year 2020. Authors need to justify this point.
Authors: Thank You for this comment. Treprostinil was first introduced in 2002 in the United States while in Poland it has been used since 2007.
Reviewer 2 Report
Comments and Suggestions for Authors
1. Do you have any information as to how long patients had had pulmonary hypertension before they were started on this medication?
2. Is it possible that the duration of disease had an effect on the escalation timeframe?
3. The mortality rate in your study seems relatively high. You might comment on that.
4. Age appears to have an important effect on functional capacity based on the six minute walk test. Do you have any more detail about the medical problems associated with increased age?
Author Response
- Do you have any information as to how long patients had had pulmonary hypertension before they were started on this medication?
Authors: Thank You for this comment. The information has been completed according to your suggestion.
In the study group of patients, the mean time from the diagnosis of PAH to the inclusion of treprostinil treatment was 2400 days (±6832), median 625 days (IQR 234-2255).
- Is it possible that the duration of disease had an effect on the escalation timeframe?
Authors: Thank You for this question. We did not explore this issue in our study, however, the escalation of the treprostinil dose at our center is mainly dependent on the patient's clinical condition and treatment tolerability.
- The mortality rate in your study seems relatively high. You might comment on that.
Authors: Thank You for this comment. The mortality rate in our study appears to be relatively high, this is most likely due to the reimbursement criteria during the study period. New reports indicate the benefit of including triple targeted therapy (PDE51 + ERA + parenteral prostacyclin) after the diagnosis of PAH (up-front therapy), while in the period studied in Poland, triple therapy was given to patients in WHO FC III or IV, in whom dual oral therapy (PDE5i + ERA) was insufficiently effective. However, there is currently no data on whether triple up-front therapy affects the mortality rate of PAH patients.
- Age appears to have an important effect on functional capacity based on the six minute walk test. Do you have any more detail about the medical problems associated with increased age?
Authors: Thank You for this question. We do not have data on comorbidities in the study group of patients, this was not analyzed in our study.